# Identification, Characterization and Synthesis of Walterospermin, a Sperm Motility Activator from the Egyptian Black Snake *Walterinnesia aegyptia* Venom

**DOI:** 10.3390/ijms21207786

**Published:** 2020-10-21

**Authors:** Tarek Mohamed Abd El-Aziz, Lucie Jaquillard, Sandrine Bourgoin-Voillard, Guillaume Martinez, Mathilde Triquigneaux, Claude Zoukimian, Stéphanie Combemale, Jean-Pascal Hograindleur, Sawsan Al Khoury, Jessica Escoffier, Sylvie Michelland, Philippe Bulet, Rémy Beroud, Michel Seve, Christophe Arnoult, Michel De Waard

**Affiliations:** 1Department of Cellular and Integrative Physiology, University of Texas Health Science Center at San Antonio, San Antonio, TX 78229-3900, USA; mohamedt1@uthscsa.edu; 2Zoology Department, Faculty of Science, Minia University, El-Minia 61519, Egypt; 3Smartox Biotechnology, 6 rue des Platanes, 38120 Saint-Egrève, France; lucie.jaquillard@smartox-biotech.com (L.J.); mathilde.triquigneaux@smartox-biotech.com (M.T.); claude.zoukimian@smartox-biotech.com (C.Z.); stephanie.combemale@smartox-biotech.com (S.C.); remy.beroud@smartox-biotech.com (R.B.); 4PROMETHEE Proteomic Platform, University Grenoble Alpes, LBFA et BEeSy, 38041 Grenoble, France; sandrine.bourgoin@univ-grenoble-alpes.fr (S.B.-V.); smichelland@chu-grenoble.fr (S.M.); michel.seve@univ-grenoble-alpes.fr (M.S.); 5PROMETHEE Proteomic Platform, INSERM U1055, 38041 Grenoble, France; 6PROMETHEE Proteomic Platform, Institut de Biologie et de Pathologie, IAB, CHU Grenoble Alpes, 38000 Grenoble, France; 7Institut Pour l’Avancée des Biosciences (IAB), INSERM U1209, CNRS UMR 5309, 38700 La Tronche, France; g-martinez@live.fr (G.M.); jp.hograindleur@gmail.com (J.-P.H.); jessica.escoffier@univ-grenoble-alpes.fr (J.E.); philippe.bulet@biopark-archamps.org (P.B.); christophe.arnoult@univ-grenoble-alpes.fr (C.A.); 8Université Grenoble Alpes, 38000 Grenoble, France; 9L’institut du thorax, INSERM, CNRS, UNIV NANTES, F-44007 Nantes, France; sawsan-khoury@hotmail.com; 10Université de Nice Sophia-Antipolis, LabEx «Ion Channels, Science & Therapeutics», F-06560 Valbonne, France; 11Archamps BioPark, 260 Avenue Marie Curie, Archamps Technopole, F-74166 Saint Julien en Genevois, France

**Keywords:** Snake venom, *Walterinnesia aegyptia*, bioactive components, sperm motility, proteomics, mass spectrometry, peptide sequencing, drug screening.

## Abstract

Animal venoms are small natural mixtures highly enriched in bioactive components. They are known to target at least two important pharmacological classes of cell surface receptors: ion channels and G protein coupled receptors. Since sperm cells express a wide variety of ion channels and membrane receptors, required for the control of cell motility and acrosome reaction, two functions that are defective in infertility issues, animal venoms should contain interesting compounds capable of modulating these two essential physiological functions. Herein, we screened for bioactive compounds from the venom of the Egyptian black snake *Walterinnesia aegyptia* (Wa) that possess the property to activate sperm motility in vitro from male mice OF1. Using RP-HPLC and cation exchange chromatography, we identified a new toxin of 6389.89 Da (termed walterospermin) that activates sperm motility. Walterospermin was *de novo* sequenced using a combination of matrix assisted laser desorption ionization time of flight mass spectrometry (MALDI-TOF/TOF MS/MS) and liquid chromatography electrospray ionization quadrupole time-of-flight mass spectrometry (LC-ESI-QTOF MS/MS) following reduction, alkylation, and enzymatic proteolytic digestion with trypsin, chymotrypsin or V8 protease. The peptide is 57 amino acid residues long and contains three disulfide bridges and was found to be identical to the previously cloned Wa Kunitz-type protease inhibitor II (Wa Kln-II) sequence. Moreover, it has strong homology with several other hitherto cloned Elapidae and Viperidae snake toxins suggesting that it belongs to a family of compounds able to regulate sperm function. The synthetic peptide shows promising activation of sperm motility from a variety of species, including humans. Its fluorescently-labelled analog predominantly marks the flagellum, a localization in agreement with a receptor that controls motility function.

## 1. Introduction

Snake venoms are complex chemical mixtures of pharmacologically active compounds such as enzymes, peptides and proteins [1,2], as well as non-protein components (carbohydrates, lipids and metal ions). Toxin peptides isolated from snake venoms target a large number of ion channels [3,4], ligand-gated channels, membrane receptors [5,6,7], enzymes [8], lipids and components of the hemostatic system [9,10] with high selectivity and affinity. Although snakebites still represent serious health concerns in tropical and subtropical countries, with more than five million cases of envenoming and between 20,000 and 100,000 deaths every year [11,12], snake venoms have also been used as medical tools for thousands of years especially in traditional Chinese medicine [13]. Actually, the latest trends in venom research demonstrate the potential of these mini natural libraries as rich sources for drug discovery [14,15,16,17,18,19,20]. In 1981, Captopril received the Food and Drug Administration (FDA) approval as the first successful example of a biomimetic drug based on a snake venom protein isolated from *Bothrops jararaca* venom. Captopril is used for the treatment of hypertension and operates by inhibiting the angiotension converting enzyme [21,22]. Tirofiban or Aggrastat, another peptidomimetic from a toxin isolated from African saw-scaled viper (*Echis carinatus*) venom, was shown to target α_IIIb_β_3_ integrin for the treatment of acute coronary syndrome and received FDA approval in 1999. Eptifibatide or Integrilin, a six-residue cyclic peptide, was first identified from the southeastern pygmy rattlesnake (*Sistrurus miliarus barbouri*) venom and also targets the α_IIIb_β_3_ integrin for the treatment of acute coronary syndrome [23,24]. It received FDA approval a year earlier in 1998. Batroxobin, a 231 amino acid protein from the lancehead snake venom that targets fibrinogen is now used outside the USA for the treatment of perioperative bleeding. Moreover, besides these approved drugs, many other snake venom components are now involved in preclinical trials for a variety of therapeutic applications [25,26]. 

With such a rich repertoire of membrane receptors targeted by venom components, all cell types should be sensitive to the action of snake venoms. Sperm cells, while not being obvious cell types targeted by animal venoms, contain a notorious rich repertoire of ion channels and membrane receptors [27], making them a priori susceptible interesting targets for the action of venom components. The sensitivity of sperm to venom components was recently demonstrated with the discovery of spermaurin from the venom of the scorpion *Maurus palmatus* [28]. In an earlier report, we have screened a snake venom bank and have shown that mice male sperm motility and acrosome reaction are inhibited by the action of venom phospholipases A2 [29]. In the present investigation, we had a very different aim and decided to screen for venom components with stimulatory functions on mice sperm motility. Most cases of sperm-linked infertilities involve defective sperm motility or altered acrosome reaction [30,31]. Sperm motility is an absolute requirement for fertilization to occur. There are several reasons for that. Fertilization is a process that happens once the sperm has travelled deep in the female track and it has to overcome several obstacles at the chemical and physical levels. This motility is also required to cross the protective layers of the oocyte and to initiate the fusion of the gametes [32]. Finally, it should be emphasized that successful fertilization also relies on the percentage of motile sperms [33], which is a reason why this parameter is so scrutinized in assisted reproductive technology. Sperm motility relies on a number of important ion channels, many being exclusively found in sperm cells [34]. These targets open new perspectives for the development of non-hormonal male contraceptive molecules. 

We therefore attempted to identify a snake venom component that may boost sperm motility function. The initial screen enabled us to identify the venom of the Egyptian black snake *Walterinnesia (W.) aegyptia* as promising sperm motility enhancer. From our extensive beforehand experience on the fractionation techniques and mass spectrometry analyses of this venom type [35], we therefore performed a screening of venom components on OF1 mice sperm motility using a first step RP-HPLC separation procedure followed by a second step cation exchange separation technique. The combination of these two techniques led to the identification of a three disulfide-bridged component of 6389.89 Da (termed walterospermin) that was subsequently *de novo* sequenced using a combination of MALDI-TOF/TOF MS/MS and LC-ESI-QTOF MS/MS techniques after reduction, alkylation and proteases-mediated peptide digestion of the isolated compound. Walterospermin is a 57 amino-acid long peptide that contains three disulfide bridges. The cDNA of walterospermin was sequenced by another group in an earlier attempt to identify *W. aegyptia* components, but the function of this peptide remained unknown [36]. This peptide has important amino acid sequence identities with several other snake venom peptides, further indicating that other animal venoms may contain toxins with sperm modulatory function properties as expected from the screening of our earlier snake venom bank [29].

## 2. Results

### 2.1. RP-HPLC Separation of W. aegyptia Venom Yields a Fraction that Positively Stimulates Sperm Motility

RP-HPLC is a method of choice for separating venom components with a good resolution (little overlap in MS ion composition among adjacent fractions) and fewer losses of components [35]. In addition, desalting steps, that result in material loss, are not required for a phenotypic screening. Separation of 2 mg of *W. aegyptia* venom by analytical C18 RP-HPLC led to 24 fractions with most venom material segregating mainly between fractions 8 and 18 (Figure 1A). All these fractions were probed against male OF1 sperm using the Computer-Assisted Semen Analysis (CASA) system to examine two motility parameters: VCL, the curvilinear velocity, and ALH, the amplitude of the lateral head displacement (Figure 1B). As illustrated, most fractions (F) induced mild alterations in VCL and ALH (F1, F3 to F10, F12, F16 to F24), four fractions produced inhibition (F2, F13 to F15), and one produced an improvement in motility parameters (F11) (Figure 1C,D). F11 increased the average VCL from 106 ± 27 µm/s (*n* = 68) to 141 ± 45 µm/s (*n* = 74) and the average AHL value from 8 ± 3 µm/s to 10 ± 4 µm/s. In agreement with these data, F11 affected the VSL (straight-line velocity) parameter as well by increasing it from 28 ± 13 µm/s to 50 ± 22 µm/s (data not shown). We focused our attention on fractions that modulated sperm motility parameters by over 20%, while control samples led to variations below 5% (data not shown).

SDS-PAGE analysis of fraction 11 of *W. aegyptia* venom indicates that most components present in this fraction have a molecular weight slightly below or above 6500 Da (Figure 2, inset). A more precise estimate of the molecular weight of the components present in F11 was obtained by MALDI-TOF MS in linear positive mode (Figure 2). Five ions were detected of *m/z* 6389.89, 6605.09, 7329.74, 7390.57, and 7605.85 ([M+1H]^1+^).

### 2.2. Cation Exchange Chromatography of F11 Leads to the Full Purification of a Bioactive Compound on Sperm Motility

Next, RP-HPLC F11 was resuspended in 300 µL of 20 mM sodium acetate buffer and separated by cation exchange chromatography. In coherence with the MALDI-TOF MS data, five major peaks were separated (from 1 to 5) using this technology (Figure 3A). Again, MALDI-TOF MS analyses of the cation exchange peaks demonstrate that, with the exception of peak 4, each peak contains a single compound with a *m/z* value that corresponds to one of the five *m/z* values detected in the RP-HPLC F11 (peak 1: *m/z* 6389.89; peak 2: *m/z* 6605.09; peak 3: *m/z* 7329.74; peak 5: *m/z* 7390.57). Peak 4 also contains a single compound but with *m/z* value of 7396, not previously detected in F11, arguably because of ionization suppression issues. None of the peaks correspond to the *m/z* 7605.85 ion probably because cation exchange is not the proper technology to isolate this compound. Next, each purified compound was challenged on sperm motility parameters VCL and ALH (Figure 3B). While compound 3 provided encouraging results, we decided to focus on compound 1 that showed greater stimulation of both VCL and ALH parameters. VCL increased from an average of 118 ± 96 µm/s (Control; N = 119; different sperm batch than for F11 analyses) to 160 ± 87 µm/s (Compound 1; N = 84), and ALH from 6 ± 5 µm/s to 8 ± 4 µm/s. VSL increased as a result from 53 ± 52 µm/s (Control) to 84 ± 53 µm/s (Compound 1). We therefore concluded that the ion species with a *m/z* value of 6389.89 was an active component of F11 that stimulates sperm motility. It was termed walterospermin.

### 2.3. De Novo Sequencing of Walterospermin by MALDI-TOF/TOF MS/MS and LC-ESI-QTOF MS/MS

We used 25 µg of purified walterospermin to perform *de novo* sequencing of this peptide by a combination of mass spectrometry techniques. MALDI-TOF MS of walterospermin indicates the presence of a major peak in positive linear mode at *m/z* 6389.89 (Figure 4A) and minor peaks corresponding to the double charge species and dimer. The reduced, alkylated, digested and ZipTip_C18_-purified peptides were analyzed by MALDI-TOF MS in reflector mode (Figure 4B). Many ions were generated by trypsin digestion such as *m/z* 930.3862^1+^, 1082.3799^1+^, 1479.5731^1+^, 1536.5903^1+^ and 3067.3298^1+^. All ions were analyzed by MALDI-TOF/TOF MS/MS and *De novo* explorer^®^ software version 3.6 (Sciex, Les Ulis, France) used to define the amino acid sequences. One sequencing example is illustrated for the precursor ion *m/z* 1536.5903^1+^ that provided a partial walterospermin 14 amino acid sequence of FMYGGCGGNANNFK (Figure 4C). 

As a result of the different ion sequencing (three in total), MALDI-TOF/TOF MS/MS covered 41 amino acids which represent 71.9% of the full sequence (Table 1).

To complete the coverage of the peptide sequencing, the same trypsin-digested samples were analyzed by LC-ESI-QTOF MS and MS/MS. The full walterospermin was also detected in different charge states (Figure 5A). Also, part of the peptide fragments resulting from trypsin digestion of walterospermin are shown on an expanded part of *m/z* ions in Figure 5B (zoom on those ions that provided the best sequencing score with Peaks^®^ Studio sequencing software V7.0; Bioinformatics Solutions Inc., Waterloo, ON, Canada ). The ions detected were *m/z* 624.8646^2+^, 688.8204^2+^, 767.6063^2+^, 1023.1354^3+^, 1375.6309^1+^, and 1534.7067^1+^. Several of these ions are common with those detected in MALDI-TOF MS. An example of *de novo* sequencing by LC-ESI-QTOF MS/MS is provided in Figure 5C for the *m/z* 1375.6309^1+^ ion. 

The resulting 11 amino acid peptide sequence was SPFYYNPHSHK. A total of 47 amino acids could be sequenced with this approach (as a result of the sequencing of four peptides), corresponding to 82.5% of the full walterospermin sequence (Table 1). This allowed the identification of a sequence of 10 more amino acids, bringing the sequence coverage to 51 amino acids out of 57. To complete the full *de novo* sequencing, walterospermin was also digested with other proteases (chymotrypsin or V8 protease). Using the same approach, chymotrypsin digestion of walterospermin provided three amino acid sequences (YNPHSHKCQKF, KTLDECHRTCVG, and KFMYGGCGGNANNFKTLDE) covering 35 residues out of 57 (64.9%). This allowed the identification of seven more amino acids of walterospermin sequence. Finally, V8 protease digestion of walterospermin provided four peptide sequences (GCGGNANNFKTLDE, RPRLCELPAE, PHSHKCQKFMYG, and CHRTCVG). This sequencing yielded the three first amino terminal residues of walterospermin. Altogether, these *de novo* sequencing efforts covered all the walterospermin sequence (Table 1).

To confirm the peptide sequencing results that were obtained after using different proteolytic digestion enzymes and two *de novo* sequencing softwares, a top-down approach was used to confirm the walterospermin sequence. In the top-down approach, walterospermin is reduced/alkylated by TCEP and iodoacetic acid, respectively, but not digested by any protease. Next, reduced/alkylated walterospermin was submitted to LC-ESI-QTOF MS and MS/MS analyses. The top-down MS provided a reduced/alkylated walterospermin *m/z* value of 6744.11 (Figure 6A). This value is to be compared to the non-reduced walterospermin *m/z* value of 6389.89, which allows to determine the number of disulfide bridges along with the number of times cysteines were alkylated. We found therefore that walterospermin contains three disulfide bridges (gain of 6 Da upon reduction) and that the reduced/alkylated walterospermin was alkylated six times (gain of 6 × 57.02 Da upon alkylation). Next, collision-induced dissociation (CID) energy was used to fragment walterospermin in the mass spectrometer by MS/MS thereby producing b and y fragment ions. The top-down spectra were analyzed manually and *de novo* assignment of the MS/MS of one 6^+^ charged ion at *m/z* 1125.0179 allowed the identification of the following 12 amino acid sequence tag LPAESGLCNAYI (Figure 6B). This sequence fully confirmed the earlier sequencing efforts by MALDI-TOF/TOF MS/MS and LC-ESI-QTOF MS/MS (Table 1).

### 2.4. Walterospermin Amino Acid Sequence, Predicted Disulfide Bridges, and Sequence Homologies with Known Toxins

The overlapping sequences of walterospermin presented in Table 1 allowed the reconstruction of the full-length walterospermin sequence (Figure 7A). The theoretical molecular weight [M+H]^+^ of walterospermin is 6395.89 Da (monoisotopic mass) for the reduced form and hence 6389.89 Da for the peptide with its three disulfide bridges which is coherent with the experimental mass of the purified peptide (Figure 5A). Walterospermin contains three-disulfide bridges (Figure 7B). MS/MS data do not make the difference between leucine and isoleucine. Nevertheless, a Blast search with walterospermin sequence showed sequence identities between walterospermin and Wa Kln-II, a cloned toxin from *W. aegyptia* venom gland [36]. This allowed the identification of three Leu and two Ile positions in walterospermin sequence. Walterospermin belongs to the family of Kunitz-type serine protease inhibitors (Figure 7A). Kunitz domains are generally peptides of about 60 amino acids in length, with three disulfide bridges and a P1 site. Functionally, they inhibit the enzymatic activity of protein degrading enzymes, such as trypsin or chymotrypsin, or both [37]. However, they have also been involved in ion channel blocking activity, such as potassium channels [38,39], blood coagulation [40], inflammation [41] or fibrinolysis [42]. Walterospermin has strong sequence identity also with Wa Kln-III toxin sequence by differing only by one amino acid at position 49 within walterospermin sequence. It is carboxylated at the end of the sequence, something that cannot be inferred by sequence cloning. It also lacks post-translational modifications. Other toxins have homologies between 60 and 79%. The 3D-structure of walterospermin was built by homology modelling (Figure 7B). Based on sequence homology, the disulfide bridge arrangement may be inferred to be C_1_-C_6_, C_2_-C_4_ and C_3_-C_5_ [36,37], but proof for this arrangement comes from an experimental investigation of this connectivity once the peptide has been synthesized and shown to be identical to its natural counterpart. 

### 2.5. Chemical Synthesis of Walterospermin

Walterospermin is present in limited amounts in the venom. Hence, for greater investigation of its pharmacological properties or simply to test some new analogs, successfully chemical synthesis is required. Walterospermin was chemically synthesized using solid phase fmoc chemistry and NCL ligation. Three peptide fragments were chemically synthesized, purified to homogeneity and ligated to each other to yield walterospermin (See experimental procedures for the details). Figure 8A–E illustrates the HPLC profiles of each synthesized peptide (P1, P2, P3), of the ligation between P1 and P2 (P1P2) and the final ligation between P1P2 and P3 to yield reduced walterospermin. The yield of NCL reaction between P1 and P2 after purification was 60%, while the yield between P1P2 and P3 was 50%. Mass spectrometry asserted that all fragments had the expected masses. Finally, the reduced walterospermin peptide was oxidized to produce walterospermin with its three-disulfide bridges (see resulting HPLC profile and proper mass in Figure 8F). The yield of oxidation was surprisingly high for an animal toxin (>40%) indicating that formation of the secondary structures and disulfide bridges were greatly facilitated by the primary structure in this particular case. The total yield of synthesis can be considered as largely above average for a challenging peptide of 57 amino acid residues containing three disulfide bridges. 

Because walterospermin possesses three disulfide bridges and that there are potentially 15 combinations possible of bridging, we first checked if the synthetic peptide has properties identical to the native peptide. For this purpose, a coelution experiment was performed on analytical RP-HPLC with the assumption that a correct disulfide bridging leads to a proper structure with same hydrophobicity properties. As shown, the synthetic walterospermin coeluted perfectly with the natural walterospermin confirming that we obtained the correct sequence and indicating that the folding of the peptide occurred properly (Figure 9A). Next, we experimentally determined the disulfide bridge arrangement of synthetic walterospermin. Theoretically, for three disulfide bridges in walterospermin, there could be 15 combinations of disulfide bridges possible, but a single one seems to occur upon folding that accounts for the bioactivity. Using a partial reduction procedure along alkylation of the free cysteine residues with a first maleimide agent (tag 1), and a complete reduction, alkylation with a second agent (tag 2). Figure 9B shows the HPLC separation of partially reduced forms after NPM alkylation. Peaks 1 and 8 were identified as folded and fully reduced peptides, respectively. Mass spectrometry analyses of other signals allowed attributing peaks 2-4 to intermediates containing two maleimide derivates with two remaining bridges, while peaks 5-7 were characterized by walterospermin with four maleimide derivates and only one remaining bridge. The detailed disulfide bridging pattern of walterospermin was elucidated by LC-ESI-QTOF MS by combining top-down and bottom-up sequencing of isolated intermediates 2–7 after full reduction and alkylation of remaining bridges in the presence of the second alkylating agent tag 2. Figure 9C shows sequence coverage obtained from intermediates 2–4 and the identification of alkylated sites by bottom-up approaches. From these recordings, the disulfide bridge arrangement was predicted to be C_1_-C_6_, C_2_-C_4_, and C_3_-C_5_ (Figure 9D).

### 2.6. Functional Evaluation of Synthetic Walterospermin

Synthetic walterospermin was evaluated for its ability to modify motility parameters of sperm from bovine and primate (human and NHP) species. These species were chosen because bovine reproduction is mainly performed with artificial insemination using frozen sperm and primate because human infertility is treated with ART using fresh and frozen sperm. For these experiments, we tested a concentration of 0.2 µM for all species. As shown, synthetic walterospermin produced a significant increase in sperm motility parameters VCL (Figure 10A and Figure 11A) and ALH in all species tested (Figure 10B and Figure 11B), a result in agreement with results shown in Figure 1 and Figure 3 with mouse sperm. Walterospermin also increased the percentage of motile and progressive sperm in bovine, whereas it was ineffective for non-primate sperm (Figure 10C,D). The percentage of motile frozen bovine sperm raised up from 27.7 ± 3.6% to 36.3 ± 5.9% (Figure 10C). In this case, the rise appeared to be due mainly from a conversion of non-motile to motile sperms (Figure 10D, Appendix A). These data indicate that the property of walterospermin to increase the velocity parameters of motile sperms is conserved across species whereas its ability to transform non-motile sperms into motile ones is restricted to bovine sperm.

As stated earlier, walterospermin belongs to the family of Kunitz-type serine protease inhibitors. The inhibitory site of action of these types of inhibitors has been mapped to the NA sequence of walterospermin (at positions 15 and 16). According to earlier work, Ala^16^ should be the most important to confer the inhibitor affinity for a given protease, while Asn^15^ should contribute to protease selectivity [43]. A disruptive mutation at position 16 should be a replacement of Ala^16^ by Trp^16^ [43]. Therefore, we similarly synthesized walterospermin_A16W_ to determine whether the effect of walterospermin was linked to protease inhibition. Walterospermin_A16W_ kept its activity of increasing the VCL and ALH parameters of frozen human sperm albeit with a slightly reduced amplitude (Figure 11A,B). These data indicate that the mode of action of walterospermin on sperm motility is unlikely to involve a protease inhibiting activity.

### 2.7. Impact of Walterospermin on Sperm Viability

A new sperm motility activator is of use only if it does not affect sperm viability. This was tested on human sperm and the viability assessed as a function of time in the absence and presence of 200 nM walterospermin (Figure 12). As shown, the synthetic peptide had no effect at all on sperm viability, a result that is coherent with the fact that increased sperm motility is hardly compatible with cell toxicity. However, the peptide did also not increase the percentage of viable cells over time, indicating that it lacks a pro-survival action as well.

### 2.8. Fluorescent Synthetic Walterospermin Labels Sperm Flagellum

While identification of the receptor of walterospermin requires extensive work using innovative reverse pharmacology approaches, we decided to detect specific labeling of walterospermin receptors at the surface of human sperm. For that purpose, we chemically synthesized walterospermin along the same lines of chemical synthesis than unlabeled walterospermin except that 4-pentynoic acid added onto the N-terminus of the P1 peptide fragment. The 4-pentynoic acid labeled P1 fragment was assembled with the P2 and P3 fragments by NCL, the full peptide was similarly oxidized and purified. Next, the Cy5 azide was grafted onto walterospermin by click chemistry to yield Cy5-walterospermin. As shown, the fluorescent analog Cy5-walterospermin specifically labels the flagellum of human sperm (Figure 13). Labelling occurs on the principal and end piece of the sperm and spares the head and mid-piece. Since the cells were not permeabilized at the time of labelling, walterospermin should act on the extracellular face of a plasma membrane receptor that is exclusively located on the flagellum. This specific distribution of walterospermin receptors is coherent with the functional effect of walterospermin on sperm motility.

## 3. Discussion

Herein, we used phenotypic screening as an approach to identify bioactive substances able to modulate spermatic function. The power of phenotypic screening over target-mediated screening is that the isolated compound has a greater chance to be active on the function that should be modified. Walterospermin could quite straightforwardly be identified through a two-step screening procedure: first, primary screening with fractions that contain, in all likelihood, over 20 compounds each, particularly for those fractions where the RP-HPLC chromatogram was enriched (F8 to F18), and second, by a secondary screening with purified compounds of the positive fractions of the primary screening procedure. The aim was to identify compounds acting on both VCL and ALH and, accordingly, synthetic walterospermin was indeed found to act on these two parameters. The success of this endeavor was expected since, in a precedent study, we came across a number of snake venom phospholipases that altered the acrosome reaction of sperm [29]. The fact that a single venom was sufficient to identify walterospermin as a compound to modulate sperm motility parameters from several species is by itself remarkable. First, it illustrates the diversity and richness of compounds a single venom may contain. Second, it exemplifies the extent to which the repertoire of ion channels and membrane receptors is enriched at the surface of sperms.

Of course, one of the drawbacks of phenotypic screening is that it does not provide a rationale information on the type of receptor that is targeted by walterospermin. We may therefore only speculate on walterospermin identity at this stage, based on the properties of the peptide. According to walterospermin sequence and structure (57 amino acids and three disulfide bridges), it belongs to the family of Kunitz-type serine protease inhibitors. As such it could act as a protease inhibitor to facilitate sperm motility. It should be noted, however, that protease inhibition has been tested in the past on this issue of mammalian sperm motility [44]. The authors found that protease inhibitors tend to block the initiation of motility, which does not argue for a positive role of protease inhibition on this function. We nevertheless tested this hypothesis by mutating the domain responsible for protease inhibition and producing walterospermin_A16W_. Despite this mutation, we preserve the effect of the peptide on sperm motility (both on VCL and ALH). This indicates that, contrary to the easiest assumption, the biological effect of the peptide does not rely on its most evident role. However, emerging evidence indicates that toxins of the type of Kunitz-type serine protease inhibitors may well be double-functional toxins, possibly also acting on ion channels [38,39]. For instance, Kunitz-type protease inhibitors, such as dendrotoxin K and I act on potassium channels [45,46]. Using a second analog of walterospermin, labelled with a fluorescent tag, the only type of information we managed to gather was the fact that walterospermin receptors are nicely localized on the principal piece of the flagellum of sperm. This is a reassuring observation since it is fully coherent with the expected function of these receptor(s) in cell motility. Among the ion channels that are localized exclusively in the principal piece of the flagellum, several are particularly interesting and relevant to motility. Candidate receptors are the following. (i) the cationic sperm channel CatSper (four isoforms known) that provides a tail-to-head Ca entry and propagation in sperm [47]. This is required to impact flagellar beat pattern and swimming behavior [48]. Point mutations in the CatSper1 gene has been involved in men infertility [49]. This channel is activated by intracellular alkalization and membrane depolarization. (ii) The Ksper potassium channel, carried by the slo3 protein. This channel is also located in the principal piece of the flagellum and is activated by membrane depolarization and membrane alkalization. It controls the resting membrane potential, mainly after capacitation [50]. However, it seems to play little role in motility in spite of its favorable distribution. (iii) Finally, the last known channel of interest may well be the Hv1, proton-selective, ion channel that regulates the intracellular pH and that also is localized in the principal piece of the flagellum [51]. It mediates the acid extrusion from sperm and should therefore be responsible for the activation of both CatSper and KSper ion channels. It might be of interest to test walterospermin as candidate modulator of these channels in the future. However, CatSper is a challenging channel to record from and requires expert electrophysiology on the sperm cell itself since it cannot be expressed in other cell types. To get a definite idea of walterospermin receptor(s) identity, it will be necessary to use reverse pharmacology, along with mass spectrometry techniques. Herein, the chemical synthesis of walterospermin is reported, using a three-fragment assembly by NCL, with a remarkable yield for a peptide of this size and complexity. The chemical syntheses of new walterospermin analogs, aimed at being used for affinity columns, may help this strategy of reverse pharmacology to be successful. This future work will be important because it will define a new sperm receptor target on which more classical molecular screening can be performed, eventually using chemical libraries that are not biologics, leading to the identification of active compounds on sperm motility that may be orally available. 

It is worth mentioning that the phenotypic screening we engaged with the snake venom opens other drug discovery perspectives. Compound 3 from the cation exchange chromatography also showed positivity in the VCL parameter indicating that it would be interesting to pursue the search of the identity of this peptide. Considering these achievements, it is reasonable to predict that this approach would allow us to identify many more compounds originating from animal venoms, provided that the primary screening is enlarged to encompass more fractionated venoms. Overall, this work adds a new component to the growing list of molecules that have been characterized from the black desert cobra *W. aegyptia* venom: actiflagelin, a first sperm modulating peptide from this venom [52], two neurotoxins [53], a phospholipase A2 [54], and acetylcholinesterase [55]. Besides providing pro-fertility compounds, the *W. aegyptia* venom also appears to possess interesting antitumor compounds: (i) its potential was established in vitro in human breast carcinoma cell lines [56] and (ii) it induces apoptosis of human prostate cancer cells [57] and mouse models of multiple myeloma [58]. These data suggest that this venom may also yield interesting new compounds in the field of cancer if appropriate phenotypic screenings are used. 

The finding that walterospermin does not affect cell viability is coherent for a compound that promotes motility. Globally, these preliminary results indicate that several technologies transfer perspectives can be envisioned for this peptide both in the veterinarian field (bovine artificial insemination) and in the field of human-assisted reproduction. For the veterinarian field, the most important parameters remain the cost of production. The excellent yield of synthesis and folding of this peptide of 57 amino-acid residues indicates that the cost of production of this large peptide can be considerably reduced. For human application, this peptide could be used to improve the rate of fertilization in vitro fertilization. Among the limits of this study, we can mention the following points. First, further experiments are needed to warrant the safety of this peptide on embryo development. Second, a complete dose-response of the peptide may provide hints about the lowest dose that can be used to promote sperm motility with the safest therapeutic index. Third, a complete structure-activity relationship should be performed by an Ala scan approach to identify all the residues of the pharmacophore. This information, combined with the experimental determination of the 3D structure of walterospermin, will help the development of peptidomimetics. Finally, of course, the identification of walterospermin receptor would open the door to new screening campaigns and the understanding of the role of this receptor in sperm motility.

## 4. Materials and Methods

### 4.1. Materials

Venom from a single Egyptian black desert snake *W. aegyptia* (Elapidae) originated from the Sinai Peninsula region of Egypt and was purchased from Alphabiotoxine Laboratory (Montroeul-au-bois, Belgium). The following materials originate from: ZipTip_C18_ pipette tips (Millipore, Molsheim, France), sperm motility (M2 medium Sigma Aldrich, Saint-Quentin-Fallavier, France), count chamber slides of 20 and 100 μm depth (Leja Products, Nieuw-Vennep, Netherlands), chymotrypsin and α-cyano-4-hydroxycinnamic acid (CHCA) mass spectrometry matrix (Sigma-Aldrich, Saint-Quentin Fallavier, France), mass spectrometry grade trypsin gold and sequencing grade V8 protease (Promega, Charbonnières-les-Bains, France).

### 4.2. RP-HPLC Fractionation of the Snake Venom

Two mg of crude *W. aegyptia* venom was fractionated into 24 fractions with an XBridge^TM^ BEH 130 analytical RP-HPLC C18 column (4.6 mm × 250 mm column of 3.5 µm) using an Agilent 1260 HPLC (Agilent Technologies, Les Ulis, France). Before injection into the RP-HPLC system, these 2 mg were dissolved in 300 μL of Buffer A (trifluoroacetic acid (TFA)/distilled H_2_O (dH_2_O) 0.1/99.9 (*v/v*)), filtered at 0.45 µm with cellulose acetate membrane (VWR International, Fontenay-sous-Bois, France). For elution of venom compounds, a 5–70% gradient of B buffer (TFA/acetonitrile (ACN)/dH_2_O 0.1/90/9.9 (*v/v/v*)) over A buffer was performed over 77 min and at a flow rate of 0.5 mL/min. 1.6 mL fractions were collected using 2.72% gradient steps. They were then lyophilized and resuspended in 100 µL distilled water for immediate or later use (–20 °C storage).

### 4.3. Cation Exchange Chromatography

Peptides present in the RP-HPLC fractions, identified as positive in the sperm motility assay, were purified by cation exchange chromatography. RP-HPLC fractions were first lyophilized, then dissolved in buffer A (300 µL, 20 mM of CH_3_COONa, pH 4.5) and peptides purified with an Agilent 1260 HPLC (Agilent technologies, Les Ulis, France) with a TOSOH Bioscience column (TSK gel SP-STAT, 7 µm, 4.6 mm ID × 10 cm L) (TOSOH Bioscience, Griesheim, Germany). Peptide elution was triggered over 30 min with a 0.5 mL/min flow rate and using a 0-1 M buffer B/A salt gradient (buffer B: 20 mM CH_3_COONa, 1 M NaCl, pH 4.5). The elution peaks were collected, lyophilized, and stored at –20 °C. In the case of motility assays with mice sperm, the major peak fractions were resuspended in 100 µL distilled water and desalted using ZipTip_C18_ pipette tips (80 µg capacity). ZipTip_C18_ desalting was performed with washing/equilibrating solution and elution solution containing 99/1/0.1 (H_2_O/ACN/TFA) and 15/85/0.1 (H_2_O/ACN/TFA), respectively.

### 4.4. Biological Samples

Human sperm was obtained from donors or patients undergoing routine semen analyses for couple infertility diagnosis or assisted reproductive technology (ART) at the fertility center of Grenoble University Hospital (Grenoble, France). Semen presenting normal parameters, according to the guidelines of the World Health Organization 2010 (World Health Organization, 2010) were used in our study. Ejaculates were liquefied at 37 °C for 30 min and sperm were washed twice in PBS 1X by centrifugation at 500× *g* for 5 min at room temperature discarding the surpernatant. After collection, the number of spermatozoa was evaluated using the counting chamber (Fast-read 102 from Biosigma, Cona, Italy). The samples were then cryopreserved and stored in the Germetheque center of biological resources (CBR with ISO-9001 and NF-S 96-900 certifications) according to a consistent procedure (AC2009-886: storage and cession authorization number). An informed consent on the use of part of their semen in research programs was signed by the patients (further approved by the CBR Germetheque scientific and ethical board). We also used sperm from 4 to 6-year-old Mauritian cynomolgus monkeys (*Macaca fascicularis*), as a model for non-human primate (NHP) sperm. These animals were euthanized thanks to other studies conducted at MIRCen (CEA/INSERM UMR1169, Fontenay-aux-Roses, France; agreement number: 92-032-02). These animals were experimented in protocols that did not interfere with spermatogenesis. Testes, with the attached epididymis, were delivered at 4 °C, and the extracted sperm from the epididymis used within 24 h following NHP death. Three NHP samples were used for this study. The frozen bovine sperm originated from IMV-Technologies (L’Aigle, France). It was kept in liquid nitrogen until use. Mice sperm were from 2 to 6-month-old OF1 male mice (Charles River, Miserey, France). The local Ethics Committee approved the scientific investigations that respected the French guidelines on the use of these animals (ComEth Grenoble N° 318, ministry agreement number #7128 UHTA-U1209-CA). Needles were used to allow sperm to escape from caudae epididymis (10 min at 37 °C in 1 mL of M2 medium), washed twice with M2 medium by centrifugation at 500× *g*, and the concentration was corrected for the CASA analyses.

### 4.5. Computer Assisted Semen Analysis (CASA)

The collected sperm were diluted in species-specific media and were added either to the “treated sample” containing the venom fraction, or the purified or synthetic peptide dissolved in ultrapure water, or to the “control sample” containing only ultrapure water. The ratios were 10 μL of compound together with 190 μL of semen sample. Treated and control samples were processed after 15 s. The delay between control and treated sperm analyses with the CASA system (Hamilton Thorn Research, Beverley, MA, USA) is below 1 min and the order of the analysis of samples were inverted between experiments, to avoid any time incubation artifacts. The sperm suspension was kept at 37 °C and introduced into an analysis chamber (Leja Products B.V., Nieuw-Vennep, Netherlands) of variable depth (100 μm for mouse sperm or 20 μm for human, NHP and bovine sperm). The following settings were used for analyses of human, NHP, bovine and mouse sperm, respectively: acquisition rate of 60; 60; 60; 60 Hz; number of frames recorded of 30; 20; 30; 45; a minimum contrast of 80; 80; 80; 50; minimum cell size of 3; 4; 5; 5; a low static-size gate of 0.85; 0.79; 0.1; 0.3; a high static-size gate of 4.24; 2.52; 3.4; 1.95; a low static-intensity gate of 0.39; 0.62; 0.3; 0.5; a high static-intensity gate of 1.4; 1.4; 1.7; 1.3; a minimum elongation gate of 0; 2; 8; 0; maximum elongation gate: 85; 50; 97; 87; a magnification factor of 1.89; 1.89; 1.89; 0.7. At least, forty motile spermatozoa were analyzed in each assay. The sperm were considered motile when the average path velocity (VAP) were > 1; 1; 1; 1. They were considered progressive when the VAP were >25; 25; 50; 30 and the path straightness (STR) were > 80; 80; 70; 70, for human, NHP, bovine and mouse sperm, respectively. The sperm motility parameters were measured at 37°C using a sperm analyzer. Two motility parameters were investigated in priority: VCL, the curvilinear velocity, and ALH, the amplitude of the lateral head displacement. VCL measures the real velocity of the sperm, whereas ALH measures the magnitude of the lateral displacement of the sperm head with regard to its average path. For the assays, we arbitrarily set positivity at –20% and +20% variations, below or above the mean values of these two velocities.

### 4.6. Measurement of Vitality of Human Sperm Treated with Walterospermin

Fresh human ejaculates (*n* = 3) were liquefied at 37 °C for 30 min and whole ejaculates were washed in 20 mL Sperm Preparation Medium (Origio, Målov, Denmark) by centrifugation at 500× *g* for 10 min at room temperature. Pellets were separated in two and resuspended in 5 mL SPM (control condition) or 5 mL SPM with 0.2 µM walterospermin (experimental condition). Thirty microliters of each condition were mixe with 30 μL of eosin 1% diluted in NaCl 0.9% and 30 μL of nigrosin 10% diluted in NaCl 0.9% every hour for six hours. Each time, sperm were layered onto a glass slide and dried. A minimum of 100 spermatozoa were analyzed for each time and condition.

### 4.7. Amino Acid Sequence Characterization

Three purification procedures of RP-HPLC followed by cation exchange chromatography were required to isolate 25 µg of walterospermin from 2 mg starting material. This was enough to attempt *de novo* sequencing by a combination of mass spectrometry techniques. The purity of walterospermin starting material was assessed by MALDI-TOF MS before sequencing was performed. The peptide was resuspended in 100 mM ammonium bicarbonate (pH 8). Next, 17 mM Tris (2-carboxyethyl) phosphine hydrochloride (TCEP, 55 °C, 1 hr) was used to reduce the peptide. Once reduction was achieved, the Cys residues were alkylated with iodoacetamide (24 mM, RT, dark, 1 hr). In this reduced/alkylated form, the peptide was digested by one of the following complementary enzymes using a 1:100 ratio (enzyme/peptide, w/w): trypsin (for MALDI-TOF/TOF MS/MS and LC-ESI-QTOF MS/MS), V8 protease (only for LC-ESI-QTOF MS/MS), or chymotrypsin (only for LC-ESI-QTOF MS/MS). Digestion occurred overnight at 37 °C and was ended by the addition of 0.1% formic acid. Resulting digested peptides were purified by ZipTip_C18_.

#### 4.7.1. MALDI-TOF MS Analyses

MALDI-TOF MS analyses were performed using a 4800 MALDI TOF/TOF™ mass spectrometer (Sciex, Les Ulis, France) in a linear positive middle mass mode (*m/z* 2000-14,000, mass tolerance 5 *m/z* units). The reduced/alkylated venom peptide was mixed with 0.5 µL of sinapinic acid at 10 mg/mL and spotted in duplicates onto MALDI Opti-TOF 384-well plate (Sciex, Les Ulis, France).

#### 4.7.2. MALDI-TOF/TOF MS/MS Analyses

MALDI-TOF/TOF MS/MS analyses were performed using a 4800 MALDI TOF/TOF™ mass spectrometer (Sciex, Les Ulis, France). The resulting digested peptides were desalted and concentrated to a 10 µL volume using ZipTip_C18_ pipette tips (3.3 µg capacity) prior to mass spectrometry analyses. Mixtures of 0.5 µL of α-cyano-4-hydroxycinnamic acid (CHCA, 10 mg/mL) and 0.5 µL of digested peptide sample were spotted in duplicates onto a MALDI Opti-TOF 384-well plate (Sciex, Les Ulis, France). The mass spectrometer was operated for both MS and MS/MS analyses in positive reflectron mode. Each spectrum was externally calibrated using the Peptide Calibration Standard II (Bruker Daltonics, Bremen, Germany). MS spectra were recorded within a mass range of *m/z* 700–4000 and a mass tolerance of 50 ppm. The MALDI-TOF/TOF mass spectrometer was set automatically to perform MS/MS focused on ions with a signal-to-noise (S/N) ratio over 10 from each MS spectrum. MS/MS spectra were obtained using collision-induced dissociation (CID) at 1 kV. MS/MS data were analyzed using DeNovo Explorer^™^ version 3.6 software (Sciex, Les Ulis, France) with the following settings: trypsin as the enzyme set; carbamidomethyl (C) as a fixed modification; and a mass tolerance at 50 ppm and 0.2 Da for MS and MS/MS data, respectively. Amino acid sequence scores between 50 and 100 were recorded. The amino acid sequence scores indicate the degree of matching between the theoretical fragmentation pattern and product ions detected in MS/MS spectra [59].

#### 4.7.3. LC-ESI-QTOF MS/MS Analyses

Acquisition of the LC-ESI-MS and LC-ESI-MS/MS data were performed using a Waters Q-TOF Xevo G2S mass spectrometer coupled to an Acquity UHPLC system with a Lockspray source. Separation of the peptide sample (5 µL) was done thanks to a BEH300 C18 column (1.7 μm, 2.1 mm ID × 150 mm L, Waters, Guyancourt, France) at a flow rate of 0.4 mL/min and with a 10–70% buffer B/A gradient over 10 min (buffer A: dH_2_O/formic acid, 99.9/0.1 (*v*/*v*); buffer B: ACN/formic acid, 99.9/0.1 (*v/v*)). Acquisition was done in the positive mode, within a *m/z* 100–2000 mass range and exploiting the Agilent MassLynx software version 4.1 (Waters, Guyancourt, France). Source parameters were set as follows: capillary voltage, 0.5 kV; cone voltage, 40 V; source temperature, 150°C; desolvation temperature, 600 °C; gas flow, 80 L/h; and gas desolvation, 1000 L/h. The MS data were sampled using a data-dependent acquisition method (DDA). In this method, the MS/MS data were acquired according to the CID activation mode that is based on candidate ion charge state and mass. The instrument is calibrated externally on *m/z* 278.1141 and 556.2771 ions at collision energy of 23 eV using a solution of leucine enkephalin eluted at a flow rate of 5 µL/min.

The PEAKS^®^ studio version 5.2 software (Bioinformatics Solutions Inc., Waterloo, ON, Canada) was used to analyze MS/MS data with these settings: trypsin, V8 protease, or chymotrypsin; carbamidomethyl (C) as a fixed modification; mass accuracy for MS/MS data at 0.05 Da; and mass accuracy for the precursor mass at 10 ppm. Scores between 50 and 100 for amino acid sequence were recorded.

#### 4.7.4. Top-Down Venomics

For the top-down technique, the LC-ESI-MS/MS experiments were carried out with the same Waters Q-TOF Xevo G2S mass spectrometer but an RP-UPLC Acquity column. Elution of the reduced/alkylated peptide sample was performed at a flow rate of 0.8 mL/min with a similar gradient as before and identical buffers. Peptide sample acquisition and analyses were performed in the positive mode using a *m/z* 50–2000 mass range for the Agilent MassLynx software version 4.1 (Waters, Guyancourt, France). Settings for MS analyses and external calibration remained unchanged. Acquisition of MS/MS data was performed by fragmenting the parent ions in the CID activation mode. The resulting MS/MS data were subject to manual interpretation.

#### 4.7.5. Database Submission

Proteomic data were submitted to the ProteomeXchange Consortium via the MassIVE Dataset Submission [60]. MALDI-TOF/TOF MS/MS data were submitted as partial datasets under the MassIVE identifier MSV000081852 (http://massive.ucsd.edu) and ProteomeXchange identifier PXD008537 (http://www.proteomexchange.org/). Similarly, the LC-ESI-QTOF MS/MS data were submitted with the following identifier: MSV000081740 (MassIVE).

#### 4.7.6. Search of Peptides Analogs

The BLAST website (http://blast.ncbi.nlm.nih.gov) was used for the search of protein similarities that had the highest sequence scorings with the identified peptides. Proteins with high similarity percentages (at least 60%) were kept from the database. The ExPASy ProtParam website (http://web.expasy.org/protparam) to provide online information about the theoretical molecular weights of the identified peptides. 

### 4.8. Chemical Synthesis of Walterospermin

Walterospermin and mutated walterospermin on protease inhibitor function (Walterospermin_A16W_) were synthesized by the chemical assembly of three peptide fragments (P1, P2 and P3 from the N-terminus to the C-terminus) using Native Chemical Ligation (NCL). The amino acid sequences of these fragments were as follows: P1 = RPRLCELPAESGL-NH-NH_2_, P2 = CNAYIPSFYYNPHSHKCQKFMYGG-NH-NH_2_ or mutated P2 = CNWYIPSFYYNPHSHKCQKFMYGG-NH-NH_2_ and P3 = CGGNANNFKTIDECHRTCVG. The three peptides were assembled stepwise using solid-phase fmoc chemistry on a Symphony Synthesizer (Protein Technologies Inc., Tucson, AZ, USA). The P1 and P2 or mutated P2 peptides were assembled using a 2-chlorotrityl chloride resin (substitution rate of 1.6 mmol/g) functionalized with hydrazine. P3 was assembled using the same resin but non-functionalized. Coupling reaction of each amino acid is 15 min (repeated three times for coupling yield increase). The peptides were cleaved from the resin and deprotected with 92.5% (vol) TFA, 2.5% H_2_O and scavengers (1,3-dimethoxybenzene (2.5%) and triisopropylsilane (2.5%). Next, they were purified by C18 RP-HPLC thanks to a Jupiter Proteo column (4 µm, 21.2 mm ID × 250 mm L) (Phenomenex, Torrance, Ca, USA) coupled to a preparative HPLC (1260 Infinity) from Agilent Technologies (Les Ulis, France). NCL reaction was used to couple P1 and P2 or mutated P2. The resulting P1P2 and P1P2 mutated peptides were purified and coupled by NCL to P3 to yield walterospermin or walterospermin_A16W_. The full-synthesized peptides were purified using RP-HPLC. Molecular masses were determined by LC-ESI-QTOF MS during syntheses of each intermediate peptides P1, P2, mutated P2, P3, P1P2, and P1P2 mutated, and for the final reduced walterospermin or walterospermin_A16W_. Finally, walterospermin and walterospermin_A16W_ were folded/oxidized in 50 mM Tris-HCl, pH 8.3 during 72 h. The resulting oxidized walterospermin and walterospermin_A16W_ with their three disulfide bridges were purified to homogeneity using RP-HPLC.

### 4.9. Determination of the Disulfide Bridge Organization of Walterospermin

To define the disulfide bridge pattern of walterospermin, a successive reduction and alkylation process was performed by using alkylating agents *N*-methyl-maleimide (NEM) and *N*-propyl-maleimide (NPM). The synthetic peptide was partially reduced with TCEP, before alkylating a first time with tag 1. Then each partially reduced/alkylated intermediate isolated was fully reduced and alkylated with the second alkylating agent tag 2. The connectivity was inferred by mass differences produced by the two alkylating agents according to LC-ESI-QTOF MS recordings by combining top-down and bottom-up approaches. The reduced/alkylated walterospermin fractions were digested by using trypsin or chymotrypsin enzymes. The enzymes were added at a 1:20 ratio (enzyme/peptide, w/w) and incubated overnight at 37 °C.

### 4.10. Sperm Staining with Fluorescent Cy5-Walterospermin

Cy5-walterospermin vial containing 0.1 mg of powder (MW: 7040 g/mol) was solubilized with water in order to obtain 0.1 mM stock solution (100×) and consequently stored at –20 °C. Semen sample (10 × 10^6^ spermatozoa/mL) was incubated with Cy5-walterospermin (1X) for 15 min at room temperature (in dark condition). After incubation, aliquots were washed in PBS 1X by centrifugation at 500× *g* for 5 min at room temperature and the supernatant discarded. Samples were then fixed with paraformaldehyde (PFA) at 4% for 45 s. Slides were next rinsed twice by dipping slide into PBS 1X for 5 min. Ten µL was layered on a poly-L-lysine-coated slide and let dry at room temperature. After leaving them completely dry, each slide was covered with 100 µL Hoechst (1:1000) diluted in PBS 1X and incubated for 5 min. The slides were mounted with DAKO fluorescence mounting medium and were kept at 4°C in the dark before confocal microscopy analyses.

### 4.11. Experimental Design and Statistical Rationale

*n* values represent biological replicate numbers (50 to 150 sperm assessed for each replicate and per condition). SigmaPlot 10.0 (Systat Software, Inc, San José, CA, USA) was used for statistical significance. Comparisons of peptide effects versus control on sperm motility were performed using paired student’s *t* tests. Mean ± SEM or SD are used for the data, as specified in the text. Two-tailed P values ≤0.05 are considered as significant. 

## 5. Conclusions

Using a single snake venom, we were able to isolate a three-finger peptide capable to boost sperm motility from several species. The full peptide sequence obtained by *de novo* sequencing was confirmed by earlier transcriptomic efforts. In spite of its consequent length, we were able to chemically reproduce this pro-fertility peptide and assess its disulfide bridge connectivity. We also report the production of a fluorescent analog of the peptide that confirms the ability of walterospermin to label sperm onto a cell compartment (the flagellum) that is fully coherent with the cell motility modulation function of the peptide. This study opens the way to the identification of a new receptor that has a pro-motility function in sperm.

## Figures and Tables

**Figure 1 ijms-21-07786-f001:**
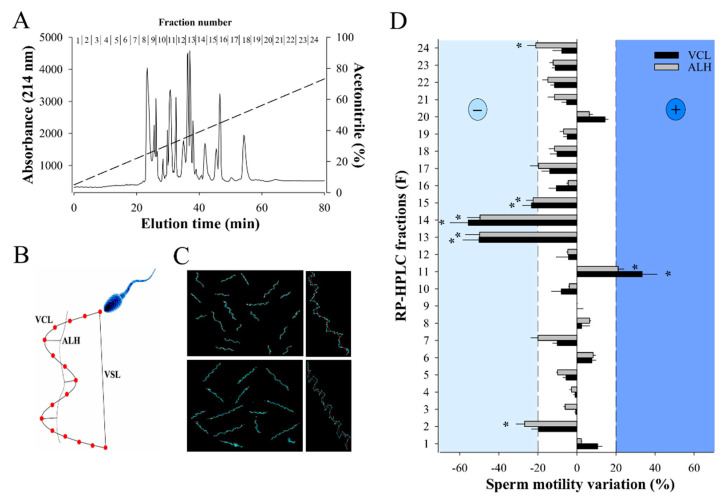
Effect of W. aegyptia venom primary fractions on OF1 mice sperm motility. (**A**) Chromatogram of the analytical C18 RP-HPLC fractionation of the Egyptian snake venom into 24 fractions. Fractions numbers are indicated on top. Dashed line represents the linear ACN gradient. (**B**) Schematic diagram illustrating the different sperm motility parameters that were calculated by the CASA system: VCL and VSL corresponding to curvilinear velocity and straight-line velocity, respectively, and ALH corresponding to the lateral displacement of sperm head. (**C**) Examples of sperm trajectories as visualized by the CASA system. Top panel: control sperms; bottom panel: 10 min F11-treated sperms. Right panels: 10× magnification on a single sperm. (**D**) Primary screening of the bioactivity of all RP-HPLC fractions on sperm motility parameters VCL and ALH. Fractions that arbitrarily produced ± 20% variation in these parameters were further considered. Results were normalized to mean variation produced by control condition. * *p* < 0.005.

**Figure 2 ijms-21-07786-f002:**
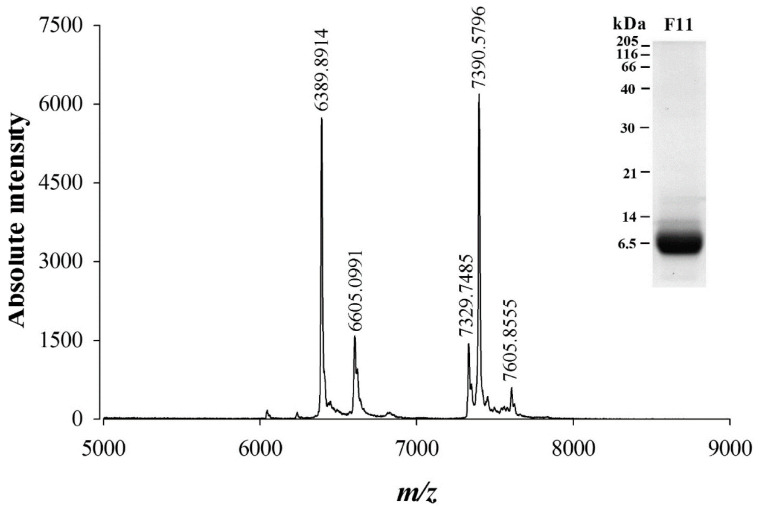
Mass characterization of F11 fraction components by MALDI-TOF MS and SDS-PAGE. Five ions were detected using the linear positive mode in the range of MW detected by SDS-PAGE migration and Coomassie blue staining (inset). [M+1H]^1+^
*m/z* values of compounds present in F11 fraction are given on top of each peak.

**Figure 3 ijms-21-07786-f003:**
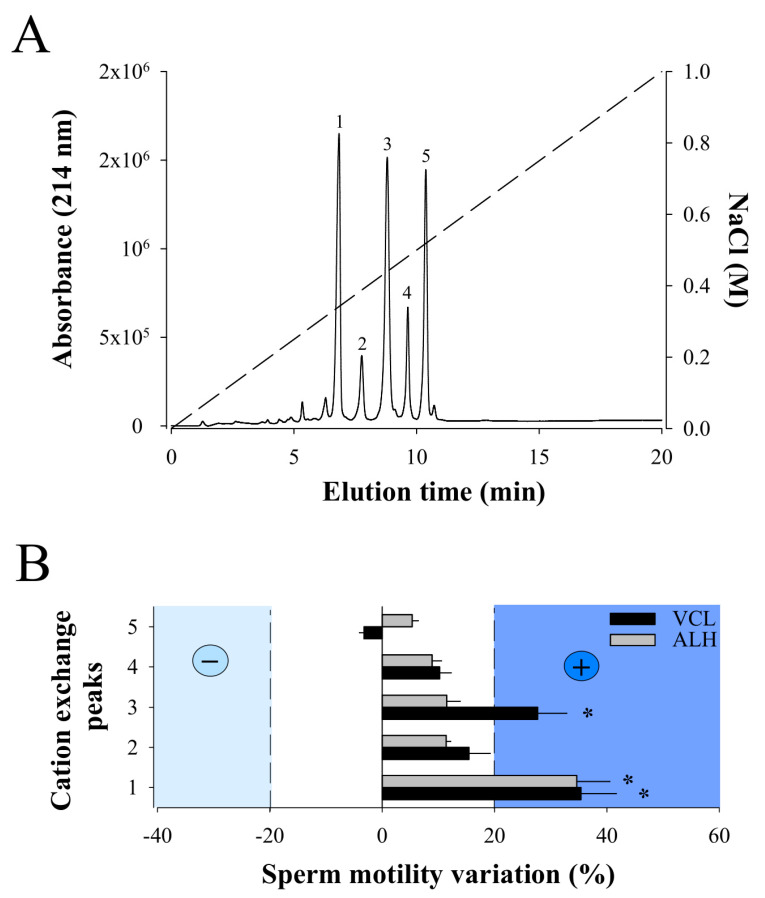
Bioactivity of purified F11 compounds. (**A**) Cation exchange chromatogram illustrating the purification of five compounds from F11 fraction. The dashed line illustrates the NaCl gradient used for elution of the compounds. Minor peaks (two before compound 1 and one after compound 5) were not purified. (**B)** Secondary screening of the compounds purified by cation chromatography on sperm motility parameters VCL and ALH. Positive compounds were considered as those that exceeded +20% variations for both VCL and ALH (compound 1). Data normalized to variation produced in control condition. * *p* < 0.005.

**Figure 4 ijms-21-07786-f004:**
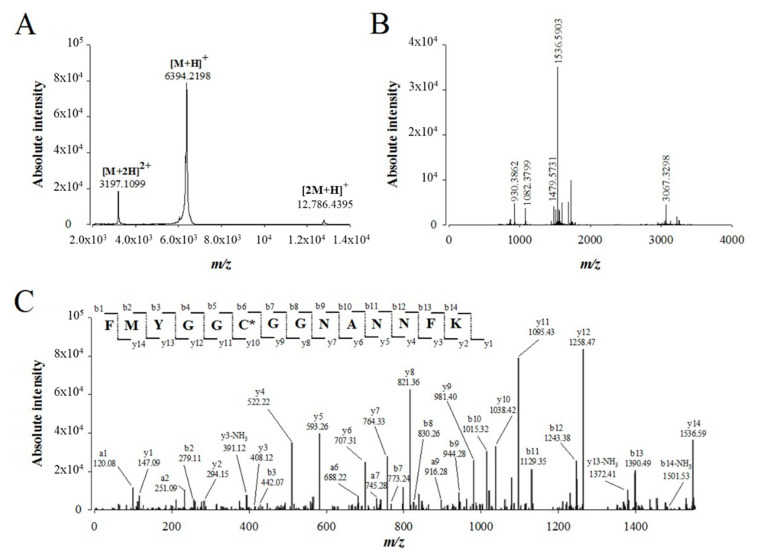
*De novo* sequencing of compound 1 from fraction F11 by MALDI-TOF/TOF MS/MS after reduction, alkylation and protease digestion. (**A**) MALDI-TOF MS of compound 1 (linear positive mode). (**B**) MALDI-TOF MS of compound 1 after reduction, alkylation and trypsin digestion (reflectron positive mode). (**C**) MALDI-TOF/TOF MS/MS of precursor ion *m/z* 1536.5903. The peptide sequence FMYGGC*GGNANNFK is derived by *de novo* sequencing. Peaks corresponding to the y, b and a ions from this peptide are labeled on the spectrum. The cysteine marked in the sequence with a star bears carboamidomethyl modification thanks to alkylation.

**Figure 5 ijms-21-07786-f005:**
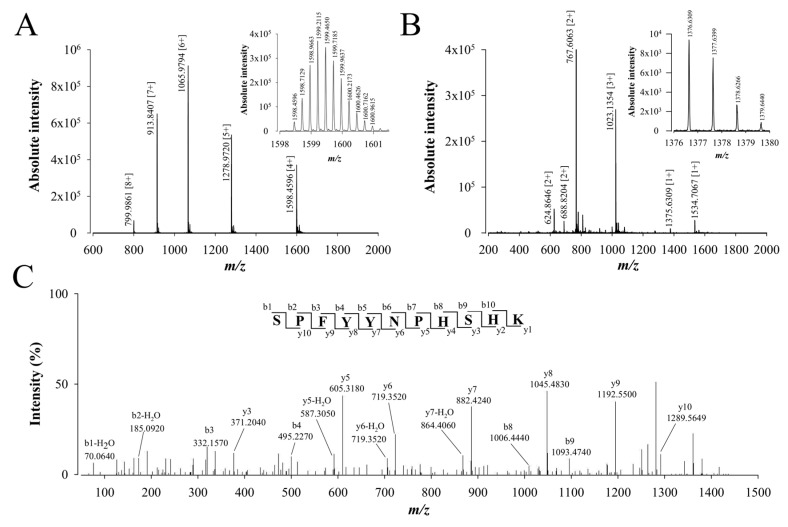
*De novo* sequencing of compound 1 from fraction F11 by LC-ESI-QTOF MS/MS after reduction, alkylation and protease digestion. (**A**) LC-ESI-QTOF MS of compound 1 (linear positive mode). Inset illustrates the MS of *m/z* 1598.4596 [4+]. (**B**) LC-ESI-QTOF MS of compound 1 after reduction, alkylation and trypsin digestion. Inset illustrates the MS of *m/z* 1375.6309 [1+]. (**C**) LC-ESI-QTOF MS/MS of precursor ion *m/z* 1375.6309. The peptide sequence SPFYYNPHSHK is derived by de novo peptide sequencing. Peaks corresponding to the y and b ions from this peptide are labeled on the spectrum.

**Figure 6 ijms-21-07786-f006:**
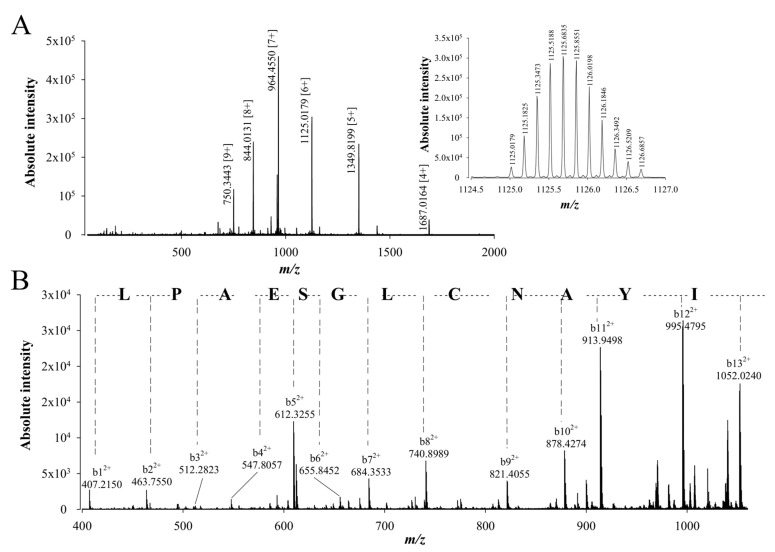
*De novo* sequencing of compound 1 from fraction F11 by LC-ESI-QTOF MS/MS after reduction and alkylation (top-down approach). (**A**) LC-ESI-QTOF MS of the reduced and alkylated compound 1 from F11. Inset: LC-ESI-QTOF MS of *m/z* 1125.0179 [6+]. (**B**) LC-ESI-QTOF MS/MS of precursor ion *m/z* 1125.0179 [6+]. The resulting peptide sequence is LPAESGLCNAYI. Peaks corresponding to the b ions from this peptide are annotated on the spectrum.

**Figure 7 ijms-21-07786-f007:**
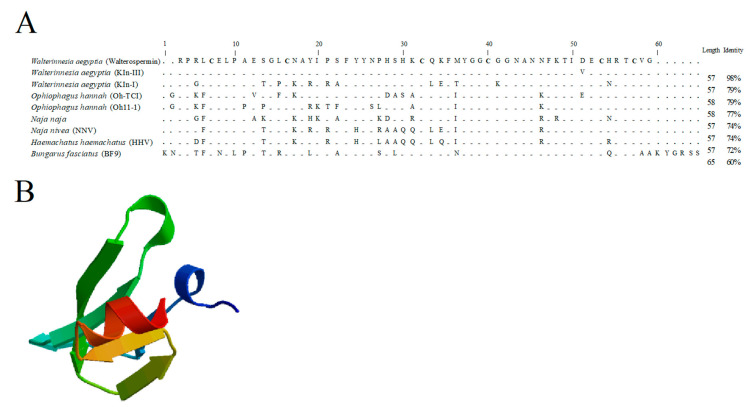
Walterospermin amino acid sequence and putative structure. (**A**) Sequence alignment of walterospermin with homolog toxins retrieved from protein BLAST. GenBank accession codes for these retrieved sequences are: C1IC51 (Kln-III), C1IC50 (Kln-I), B6RLX2 (Oh-TCI), P82966 (Oh11-1), P20229 (NN), P00986 (NNV), P00985 (HHV) and P25660 (BF9). Hyphen-minus represents identical amino acid residues, and dots indicate the lack of residue at the position. The peptide lengths and percentages of sequence identities are given on the right. (**B**) SWISS-MODEL (http://swissmodel.expasy.org/) proposed 3D-structure of walterospermin. Red, α helix; green and yellow, β-strands; blue, N-terminal.

**Figure 8 ijms-21-07786-f008:**
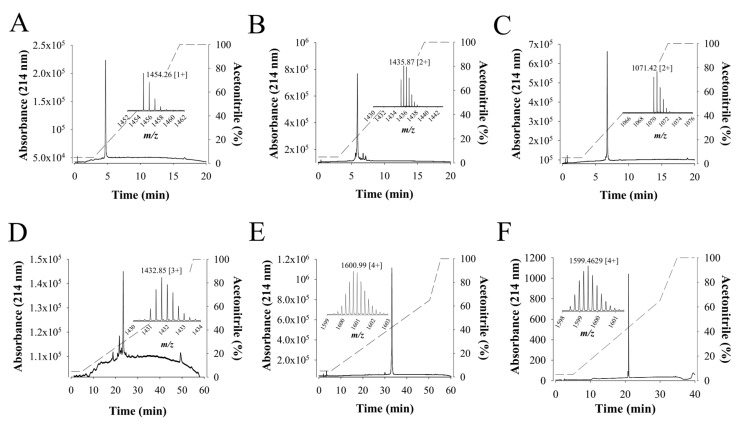
Chemical synthesis of walterospermin. (**A**) RP-HPLC purification of peptide fragment P1. Inset, LC-ESI-QTOF MS of *m/z* 1454.26 [1+]. The acetonitrile gradient is provided in dashed lines: 5 to 65% for 10 min and 65 to 100% for 2 min. (**B**) RP-HPLC purification of peptide fragment P2. Inset, LC-ESI-QTOF MS of *m/z* 1435.87 [2+]. The acetonitrile gradient as in A. (**C**) RP-HPLC purification of peptide fragment P3. Inset, LC-ESI-QTOF MS of m/z 1071.42 [2+]. The acetonitrile gradient as in A. (**D**) RP-HPLC purification of P1P2 after NCL. Inset, LC-ESI-QTOF MS of *m/z* 1432.85 [3+]. Acetonitrile gradient is provided in dashed lines: 5 to 65% for 10 min and 65 to 100% for 2 min. (**E**) RP-HPLC purification of non-oxidized walterospermin (P1P2P3). Inset, LC-ESI-QTOF MS of *m/z* 1600.99 [4+]. The acetonitrile gradient is provided in dashed lines: 5 to 65% for 50 min and 65 to 100% for 3 min. (**F**) RP-HPLC purification of oxidized walterospermin. Inset, LC-ESI-QTOF MS of *m/z* 1599.46 [4+]. The acetonitrile gradient is provided in dashed lines: 5 to 65% for 25 min and 65 to 100% for 5 min.

**Figure 9 ijms-21-07786-f009:**
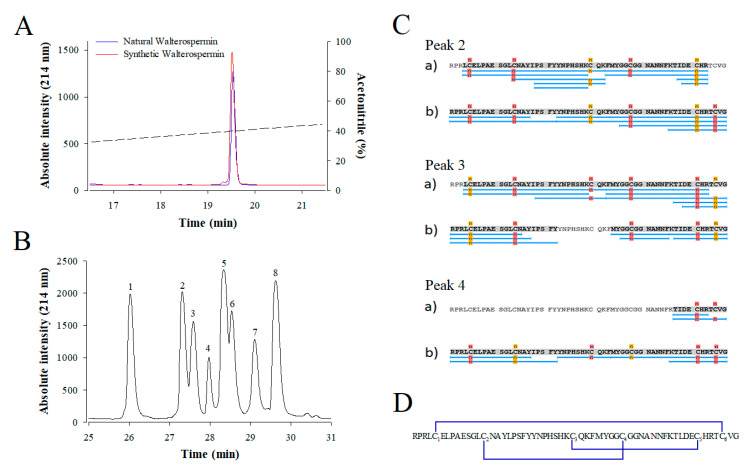
Coelution of synthetic and natural walterospermin and disulfide bridge assignment. (**A**) Coelution experiment. Natural (blue) and synthetic (red) walterospermin elution profiles are shown after separation on analytical RP-HPLC (AdvanceBio Peptide Map 2.1 × 250 mm, 2.7 µm from Agilent Technologies). The ACN gradient is shown as dashed line. (**B**) HPLC-UV (214 nm) trace of walterospermin after partial reduction and alkylation. Peak 1: folded walterospermin; peak 2-4: walterospermin with one reduced/alkylated bridge; peak 5–7: walterospermin with two reduced/alkylated bridge; peak 8: fully reduced/alkylated peptides. (**C**) Sequence coverage obtained by LC-ESI-MS(/MS) analyses of (a) tryptic and (b) chymotryptic digests of intermediates 2-4 containing N-propyl-maleimide (NPM) and N-ethyl-maleimide (NEM). Blue lines indicate peptide sequences analyzed. Yellow and red labels indicate NPM and NEM derivates, respectively. (**D**) Disulfide bridge arrangement of synthetic walterospermin. Cysteines were numbered by their order of appearance in the sequence. Underlined sequence corresponds to the active site of Kunitz-type protease inhibitors.

**Figure 10 ijms-21-07786-f010:**
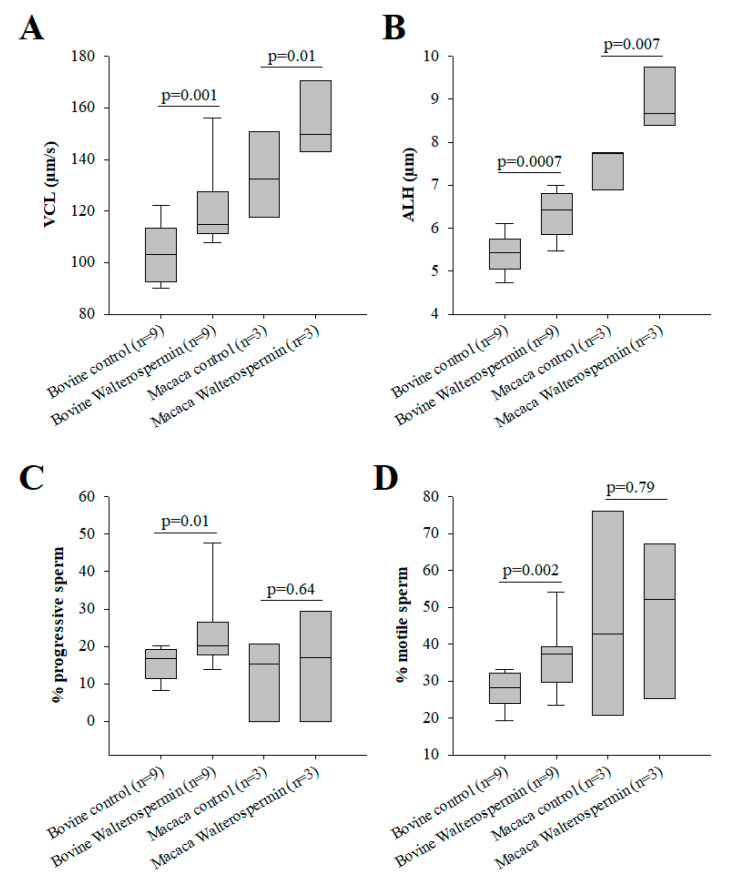
Bioactivity of synthetic walterospermin. (**A**) Increase in bovine and macaca sperm motility parameter VCL induced by 0.2 µM walterospermin. (**B**) Increase in bovine and macaca sperm motility parameter ALH induced by 0.2 µM walterospermin. (**C**) Effect of synthetic walterospermin on the percentage of motile sperm (non-progressive and progressive sperm combined). (**D**) Effect of synthetic walterospermin on the percentage of progressive sperm.

**Figure 11 ijms-21-07786-f011:**
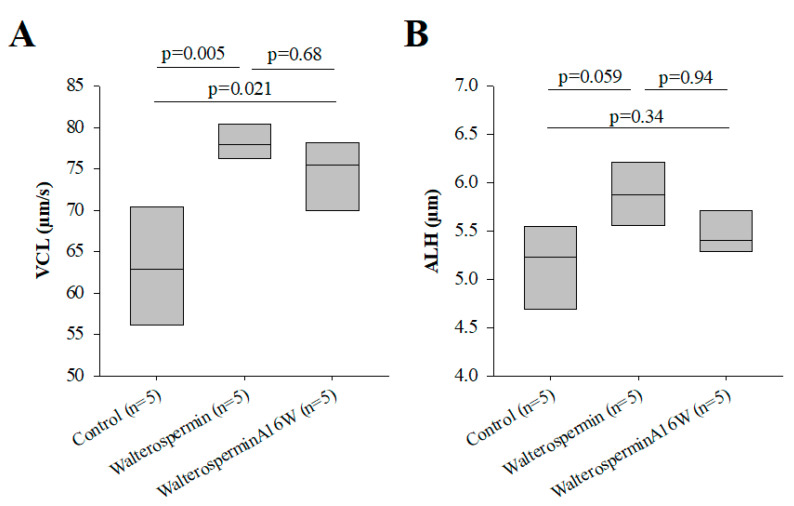
Mutation leading to the loss of protease activity (A16W) does modify the activity of the peptide. (**A**) Increase by 0.2 µM of VCL motility parameters in human sperm of frozen origin. (**B**) As in (A) but for ALH parameter.

**Figure 12 ijms-21-07786-f012:**
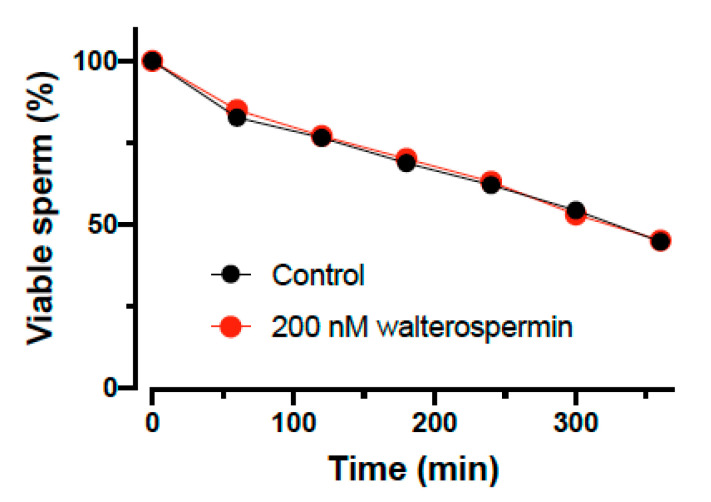
Percentage of viable human sperm as a function of time in the absence or presence of 200 nM walterospermin. The data were normalized to the initial level of survival of human sperm at t = 0 min.

**Figure 13 ijms-21-07786-f013:**
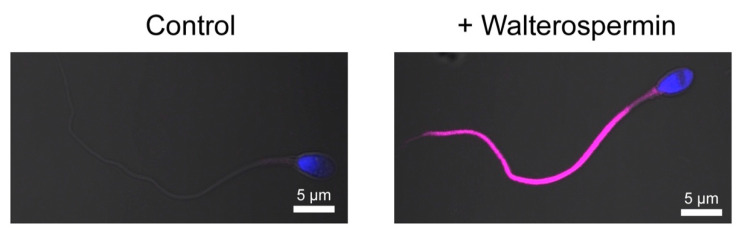
Specific labeling of human sperm flagellum by the fluorescent analog Cy5-walterospermin. Control sperm (**left**) and sperm incubated with fluorescent Cy5-walterospermin (**right**) were fixed and fluorescence was assessed by confocal microscopy. Fluorescent walterospermin signal was found to be located at the surface of the flagellum and not on the head.

**Table 1 ijms-21-07786-t001:** List of walterospermin sequences obtained after MS/MS analyses of the reduced/alkylated/digested peptide. Alkylation is witnessed by an additional mass of 57.02 Da.

Precursor Ion (*m/z*)	Charge Number (z)	MS/MS-Derived Sequence	Protease Used & Mode of Measurement
1536.60	1	FMYGGC(+57.02)GGNANNFK	Trypsin - MALDI- TOF/TOF MS/MS
930.39	1	TIDEC(+57.02)HR	
2384.07	1	ESGLC(+57.02)EKGFPSFYYNPHSHK	
768.82	2	FMYGGC(+57.02)GGNANNFK	Trypsin - LC-ESI-QTOF MS/MS
465.71	2	TIDEC(+57.02)HR	
688.82	2	SPFYYNPHSHK	
570.26	3	LC(+57.02)ELPAESGLC(+57.02)NAYI	
723.34	2	YNPHSHKC(+57.02)QKF	Chymotrypsin - LC-ESI-QTOF MS/MS
738.34	2	KTIDECHRTC(+57.02)VG	
707.64	3	KFMYGGC(+57.02)GGNANNFKTIDE	
620.82	2	RPRLC(+57.02)ELPAE	V8 protease - LC-ESI-QTOF MS/MS
749.32	2	GC(+57.02)GGNANNFKTIDE	
76.35	2	PHSHKC(+57.02)QKFMYG	
445.19	2	C(+57.02)HRTC(+57.02)VG	

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
