# Peer review of "Identification, Characterization and Synthesis of Walterospermin, a Sperm Motility Activator from the Egyptian Black Snake Walterinnesia aegyptia Venom"

_ijms, 2020, doi:10.3390/ijms21207786_

Round 1

Reviewer 1 Report

This study aimed to screen for bioactive compounds from the venom of the Egyptian black snake Walterinnesia aegyptia.

Results show that one of the compounds present in the venom, termed spermatin, induced an alteration on sperm flagellar motility. Authors also provided evidence that the modified compound, the fluorescent analogue Cy5-spermatin, was able to label the sperm cell tail.t

However, isolation of one among the venom compounds is only the starting point and not the aim of a study. The ¾ of the manuscript is focused on the description of the chromatographic techniques, with no critical description/investigations of the real physiological effects of this compound (previously isolated and called by a different name, in 32 and 33) on sperm cells. What is the target of this “spermatin”? Membrane? Ion channels? Acrosome reaction?

Many other concerns arise from the manuscript:

  1. How was this terming “spermatin” from? If authors choose it arbitrarily, they have to identify it with much more precision to include it in the protein/compound database, taking also into account that the term spermatin is already present to identify a substance similar to serumalbumin in seminal fluid.
  2. Most part of the manuscript is a mere description of the analytical methods (MALDI51 TOF/TOF MS/MS and LC-ESI-QTOF MS/MS) applied, and should be included in the methods.
  3. Discussion is insufficient, because the authors once more described results without integrating and developing the biological impact of their study. Moreover,  this compound was previously identified by the same group (Wa Kln-II), thus excluding the novelty.
  4. The mere CASA analysis for the evaluation of sperm parameters is very limited and risks to lead to misleading results, due to the low reliability of the test.

Only some of many formal mistakes are highlighted in the enclosed pdf with comments.

Author Response

This study aimed to screen for bioactive compounds from the venom of the Egyptian black snake Walterinnesia aegyptia.

Results show that one of the compounds present in the venom, termed spermatin, induced an alteration on sperm flagellar motility. Authors also provided evidence that the modified compound, the fluorescent analogue Cy5-spermatin, was able to label the sperm cell tail.

However, isolation of one among the venom compounds is only the starting point and not the aim of a study. The ¾ of the manuscript is focused on the description of the chromatographic techniques, with no critical description/investigations of the real physiological effects of this compound (previously isolated and called by a different name, in 32 and 33) on sperm cells. What is the target of this “spermatin”? Membrane? Ion channels? Acrosome reaction?

Response: Performing a screening process until completion is not a simple task: it comprises a valid functional assay for screening, purification of the compound, identification through de novo sequencing and confirmation of the biological activity upon chemical synthesis. The mere fact that this compound was cloned before (without any indication of its activity) does not affect the novelty of our findings. Chemical synthesis of such a large peptide is particularly challenging. It required a strategic approach here. The fact that the chemical counterpart is identical and as active as the native one is also a very good information that these studies can be pursued without having to refer to the purified compound from venom. The simple chemical synthesis of this peptide could represent a stand-alone manuscript. The assignment of disulfide bridges of such a large peptide is almost never performed. Here the assignment has been done without ambiguity. Among other novelties is the fact that the protease inhibitor activity of the peptide is not involved in sperm activation. This also was not inferred from the cloning of this peptide. Finally, we managed to chemically synthesize a fluorescent analogue of this peptide. We invite the reviewer to list the number of toxins that have been produced with fluorescent tags. They can be counted with the fingers of the hand. We probably did not develop enough these arguments in the manuscript. We tried to correct this issue in the revised version. Now of course, we understand the frustration of the reviewer. We also would like to understand which sperm receptor is able to bind our peptide and trigger sperm response. This information is of great interest in the future to develop small compounds as peptidomimetics acting as sperm activators. To reach this goal, a whole new set of syntheses and biochemical/proteomic analyses are needed that go way beyond the scope of this first manuscript where the identification of a biological compound acting for sperm stimulation is a first promising step. According to the structure of this peptide, there is little chance that this peptide acts on an ion channel. GPCR as a target present on the membrane of the flagellum cannot be excluded since it is a three-finger toxin (this was proposed by the way in our conclusion of the first version of the manuscript). Acrosome reaction would require a different localization. Peptides of this size and nature rarely act on ion channels. Hence, the peptide should normally act on an unknown receptor, probably a GPCR, present at the membrane of the flagellum.

Many other concerns arise from the manuscript:

  1. How was this terming “spermatin” from? If authors choose it arbitrarily, they have to identify it with much more precision to include it in the protein/compound database, taking also into account that the term spermatin is already present to identify a substance similar to serum albumin in seminal fluid.

Response: At the time we wrote this manuscript the term « spermatin » was not used. We delayed in the submission of this work and since then the term has appeared. It seems wise to change it indeed. This is what we have done. It was replaced by the term « Walterospermin » throughout the manuscript, a new term that indicates that it is a compound from Waterinnesia venom acting on sperm.

  1. Most part of the manuscript is a mere description of the analytical methods (MALDI51 TOF/TOF MS/MS and LC-ESI-QTOF MS/MS) applied, and should be included in the methods.

Response: There was indeed quite some description of the methods and their rationale in the result section. We have reorganized these paragraphs and better detailed the Methods section itself. We thank the reviewer for this suggestion.

  1. Discussion is insufficient, because the authors once more described results without integrating and developing the biological impact of their study. Moreover, this compound was previously identified by the same group (Wa Kln-II), thus excluding the novelty.

Response: we discussed this aspect of novelty here above. If all cloned sequences were preventing new findings with the same sequence, then indeed it would be sufficient to sequence the genome of an organism and nobody could claim any new novelty. Fortunately, this is not the case and DNA sequences have been excluded from patentability, illustrating how little invention rests behind a cDNA sequence. We now tried to better integrate and develop the biological impact of our findings in the discussion and thank the reviewer for this suggestion.

  1. The mere CASA analysis for the evaluation of sperm parameters is very limited and risks to lead to misleading results, due to the low reliability of the test.

Response: The CASA analysis has been solid enough to orient and select a toxin that in the end, after chemical synthesis, is both able to regulate the activity of sperm (again evaluated by CASA indeed) but also bind onto the flagellum of sperm. During the screening procedure, we are not aiming for a full description of the effects of venom fractions on sperm properties, but we aim at identifying a new ligand that will help us ultimately to identify a new important sperm receptor for motility function.

In addition, the reliability of the CASA system depends greatly of the imaging hardware and software, the instrument settings, the quality of analysis chamber and the experience of the technician. This work was performed with the latest generation of CASA system from Hamilton Thorn (CEROS II-2017), with amazing features, state of the art imaging chamber (Leija, NL) and a very skilled technician. Using a similar strategy, we have already identified biological compounds able to reproducibly inhibit or activate sperm motility (Escoffier, et al, Journal of Cell Physiology 2011, Martinez et al, Molecular human reproduction 2017, for instance). Moreover, we did not observe a difference between two measures made from control samples. Altogether, we are confident that our results are reliable.

Only some of many formal mistakes are highlighted in the enclosed pdf with comments.

Response: We thank the reviewer for highlighting these issues. We corrected them all. We shall forward this version back with responses to comments on PDF to the Editors.

Reviewer 2 Report

Abd El-Aziz et al. present a study wherein a venom-derived peptide, Spermatin, was isolated from Walterinnesia aegyptia.  The rationale for the study was to determine if peptides isolated from such venom known to affect a variety of receptors and ion channels could affect sperm motility.  The authors originally used murine sperm but also include a dataset with human sperm.

The authors present a meticulous, multi-step work that involved isolation of peptides from venom that were found to enhance motility, mass spec analysis to sequence the peptide, three-dimensional reconstruction of the peptide, manufacture of the peptide, verification that the manufactured peptide enhanced sperm motility, and finally immunolocation experiments to localize the attachment of the peptide to within the sperm flagellum.

The experimental design and methods are excellent, and the data support the conclusions drawn by the authors.  I have no important criticisms of the work.

I am puzzled by the concluding statements of the Discussion section.  The authors note that production of this new peptide will likely be less costly in the future, but will likely have veterinary rather than human application.  I assume that they mean to improve the efficiency of fertilization in animals.  Why not humans, especially in the case of male infertility secondary to poor motility.  The authors need to expand on this point.

Author Response

Abd El-Aziz et al. present a study wherein a venom-derived peptide, Spermatin, was isolated from Walterinnesia aegyptia.  The rationale for the study was to determine if peptides isolated from such venom known to affect a variety of receptors and ion channels could affect sperm motility.  The authors originally used murine sperm but also include a dataset with human sperm.

The authors present a meticulous, multi-step work that involved isolation of peptides from venom that were found to enhance motility, mass spec analysis to sequence the peptide, three-dimensional reconstruction of the peptide, manufacture of the peptide, verification that the manufactured peptide enhanced sperm motility, and finally immunolocation experiments to localize the attachment of the peptide to within the sperm flagellum.

The experimental design and methods are excellent, and the data support the conclusions drawn by the authors.  I have no important criticisms of the work.

Response: we thank the reviewer for this appreciation of the work.

I am puzzled by the concluding statements of the Discussion section.  The authors note that production of this new peptide will likely be less costly in the future, but will likely have veterinary rather than human application.  I assume that they mean to improve the efficiency of fertilization in animals.  Why not humans, especially in the case of male infertility secondary to poor motility.  The authors need to expand on this point.

Response: the reviewer is right. We now expanded this discussion on the potential use of the peptide for male infertility. See conclusion.

Reviewer 3 Report

It's regret that the it is not sufficient data to show the identified peptides’ characterization. Besides, I have some concerns below:
1. The plagiarism is 42%, suggesting revising the manuscript by using authors own words so that avoiding the higher plagiarism.

2. Author should provide the sample preparation for RP-HPLC, different lysate buffer will give huge different results.

3.it is better if authors add a process flowchart to isolate/purify the venom and have the table to indicate the purity in each step.

4. How did author repeat the experiments either using same batch of samples or different batch of samples?

5. Does MAILD-TOF/TOF/MS/MS and LC-ESI-QTOF MS/MS directly sequence whole peptides? If so, why do you employ different ways to get whole sequences? Could you present the individual whole sequences if you can get whole sequence separately?

6. did the author splice the full sequence of compound1 from fraction 11? I again wonder if MALDI-TOF/TOF MS/MS or LC-ESI-QTOF MS/MS can be used to get full sequences? if so, why utilizing two ways? if not, how the methods can complementary in the sequences of proteins?

7. to let readers clearly understand the manuscript, I do think the author should be introducing the advantages/disadvantages/application for used different MS/MS and digested ways.

8. what is the differences between using TCEP and iodoacetic acid and protease?

Author Response

It's regret that the it is not sufficient data to show the identified peptides’ characterization. Besides, I have some concerns below:

  1. The plagiarism is 42%, suggesting revising the manuscript by using authors own words so that avoiding the higher plagiarism.

Response: Always difficult to reinvent your own text in the Methods section. Nevertheless, the incriminated paragraphs were highlighted and we rephrased them all in order to avoid this observation. We hope we satisfactorily addressed this point.

  1. Author should provide the sample preparation for RP-HPLC, different lysate buffer will give huge different results.

Response: For RP-HPLC, we do not use lysates. We use resuspended venom in water (it never varies in composition) or we use resuspended peptide in water too. There again there is no reason to expect any variability and different results. The same peptide passed ten times on the same column and in the same conditions will always give the same results.

3.it is better if authors add a process flowchart to isolate/purify the venom and have the table to indicate the purity in each step.

Response: We would be happy to add a flowchart but it is quite basic: a first fractionation of the venom giving 24 fractions (illustrated in Fig. 1) but where we cannot speak about purity, and a second purification step with cation exchange providing 100% purity.

  1. How did author repeat the experiments either using same batch of samples or different batch of samples?

Response: We repeated the purification of walterospermin 3-6 times, we also get the same result. But this is logic since it is the same starting material. This question would be relevant if we used different batches of venoms coming from different animals. Here, we use a single animal because it was difficult to locate. If we had used different animals, then indeed the relative quantities of walterospermin in the venom and the venom composition would differ slightly depending on region of catch, diet of the animal or sex of the animal.

  1. Does MAILD-TOF/TOF/MS/MS and LC-ESI-QTOF MS/MS directly sequence whole peptides? If so, why do you employ different ways to get whole sequences? Could you present the individual whole sequences if you can get whole sequence separately?

Response: No, none of these techniques allow for a complete sequencing of the whole sequence of walterospermin. We have to reduce the peptide, alkylate it, and digest it in small pieces. Only the small peptides can be sequenced. Then it is like a puzzle and you need to determine the position of each small sequence in the large whole sequence. This is why it is better to come with many different approaches to produce overlapping sequences that help reconstitute the puzzle.

  1. did the author splice the full sequence of compound1 from fraction 11? I again wonder if MALDI-TOF/TOF MS/MS or LC-ESI-QTOF MS/MS can be used to get full sequences? if so, why utilizing two ways? if not, how the methods can complementary in the sequences of proteins?

Response: Yes, we splice them. No, the sequence of compound 1 is too large for a full sequencing by MALDI TOF/TOF or LC-ESI-QTOF.

  1. to let readers clearly understand the manuscript, I do think the author should be introducing the advantages/disadvantages/application for used different MS/MS and digested ways.

Response: MALDI is less complete than LC-ESI-QTOF because some digested peptides do not desorb easily from the matrix. But these considerations are very classical in the literature and experts in the field would not appreciate that we go into such details. If really needed, we would be happy to do it, but in our opinion, it falls outside the scope of this study.

  1. what is the differences between using TCEP and iodoacetic acid and protease?

Response: As stated in the manuscript, TCEP reduces the disulfide bridges (we now precise this point in the text), iodoacetic acid adds an alkylating agent onto the liberated Cysteine residues (to avoid that they again form disulfide bridges) and the proteases cut walterospermin in smaller pieces to help sequence determination.

Round 2

Reviewer 1 Report

The authors provided many changes to the manuscript, but it still needs major revisions

Introduction

  1. Authors should focus on the aim of the study and rewrite the whole chapter, since introduction is only describing venoms and their properties. Only few mentions about sperm motility, flagellar movement mechanism and its regulation has been included. The “making them a priori susceptible interesting targets for the action of venom components” is left without any explanation, for instance.
  2. “The peptide is 57 amino acid residues long and contains three disulfide bridges and was found identical to the previously cloned 54 Wa Kln-II sequence” was deleted in the revised version but it should be included describing also some features of the protease inhibitor family, whose walterspermin seems to be part.

Results

To understand the role of this compound, a dose-dependent effect should be clearly performed not only for the purified and/ or “de novo” synthetized peptide, but also for the starting sample of venom. Is this effect regulated by a curve, or is it linear? We are talking about venom, and it is reasonable that side effects may seriously affect the cell.

This leads to the second important issue: viability. The addition of this compound can affect also sperm survival and or functioning. Authors can’t assess simply that “preliminary results” show no alterations in sperm viability, they must show that.

Discussion

All this part is still describing the methods. It is expected that authors describe the main peptide properties, which can also be deduced properly from the techniques applied for its identification and "de novo" synthesis (ionic strength, disulfides, etc.). Moreover, authors should integrate this information in the wider background referring to sperm motility, since many compounds have been described to affect sperm motility, from viruses’ capside proteins to natural compounds. Is this peptide expected to affect ion channels, mitochondria related metabolism, whatsoever else-…..

No English revision has been actually reported also if repeatedly requested

Other comments have been inserted throughout the pdf version

Author Response

C1. The authors provided many changes to the manuscript, but it still needs major revisions.

Response: We appreciate the efforts made by the reviewer to improve the quality of our manuscript. We completed already a great number of modifications in response to all initial comments. While this contributes to manuscript advance, it would be comfortable to receive the entire set of comments in the first review. Nevertheless, we did our best to satisfy the reviewer on these new requests.

--------------------------------

C2. Introduction

  1. Authors should focus on the aim of the study and rewrite the whole chapter, since introduction is only describing venoms and their properties. Only few mentions about sperm motility, flagellar movement mechanism and its regulation has been included. The “making them a priori susceptible interesting targets for the action of venom components” is left without any explanation, for instance.

Response: We did not rewrite the entire manuscript introduction because the parts that we were describing are important for the toxinology readership. We now added a paragraph describing the importance of sperm motility for sperm physiology and indicate that the role that ion channels play in the coordinated movement of the flagellum. The reviewer will notice that we considerably rewrote the discussion in agreement with the comments made on this part.

--------------------------------

C3.

  1. “The peptide is 57 amino acid residues long and contains three disulfide bridges and was found identical to the previously cloned 54 Wa Kln-II sequence” was deleted in the revised version but it should be included describing also some features of the protease inhibitor family, whose walterspermin seems to be part.

Response: We are not sure to understand what the reviewer means by “was deleted in the revised version”. It was never deleted and is clearly present in the abstract of all submitted versions. Concerning some description of the protease inhibitor family, we now added several sentences at end of paragraph 2.4 (highlighted in yellow).

--------------------------------

C4. Results

To understand the role of this compound, a dose-dependent effect should be clearly performed not only for the purified and/ or “de novo” synthetized peptide, but also for the starting sample of venom. Is this effect regulated by a curve, or is it linear? We are talking about venom, and it is reasonable that side effects may seriously affect the cell.

Response: We do not know if the reviewer is familiar with venoms, but a dose-response curve with a venom is meaningless.

First, testing a whole venom blunts all effects. We did not even try to do that experiment in our project for the following reasons: a snake venom contains both agonist and antagonist compounds on sperm motility. Adding the whole venom provides a global non-interpretable result. This is why it is fractionated, to simplify the number of compounds and decomplexify the response. Also, snake venoms contain a number of cytotoxic enzymes that lead to cell death and lipid digestion. There is thus no valuable response to be expected from applying a whole venom on sperm. This is generally true, and we never test whole venoms when searching for an effect.

Second, venoms are complex mixtures of compounds, all coming in variable natural abundance and concentrations. By performing a dose-response of a venom, we would not only observe a response that is fully incoherent with sperm motility increase, but we would also have absolutely no clue about the active concentration of the compound. This is also true for the primary fraction of the venom or even the secondary one: the initial concentration of the active compound is unknown. Dose-responses of unknown concentrations are meaningless.

Third, comparing native and synthetic walterospermin is also not of any valuable interest since we have demonstrated, without any ambiguity, that both compounds are fully identical. What is the point of comparing identical products? This is why the synthesis of the compound is important to start with. In the absence of further knowledge of the receptor of walterospermin, a dose-response curve will not add any precious information on the nature of the receptor. We illustrate potent activity at 200 nM. This indicates that walterospermin binds with a quite high affinity onto its receptor(s), but the dose-response curve (that should be sigmoidal and not linear) will add no more information, apart from considerably delaying the publication of this manuscript.

Finally, we ran out of the synthetic peptide during my move from the Grenoble neuroscience institute to the institute du thorax. Re-synthesizing it would take 2 more months. Performing these experiments would take a further 2 months. By itself, this is not the best argument for not performing the suggested experiment, but we are convinced that it doesn’t bring any novelty to our findings. We will therefore not perform this experiment.

--------------------------------

C5. This leads to the second important issue: viability. The addition of this compound can affect also sperm survival and or functioning. Authors can’t assess simply that “preliminary results” show no alterations in sperm viability, they must show that.

Response: We have done this experiment with what was left of peptide in our freezers. As shown now, walterospermin has no effect on human sperm viability. This was quite evident to us as good motility is a sign of good viability of a sperm. This is now confirmed by the data. We added these data in the manuscript.

--------------------------------

C6. Discussion

All this part is still describing the methods. It is expected that authors describe the main peptide properties, which can also be deduced properly from the techniques applied for its identification and "de novo" synthesis (ionic strength, disulfides, etc.). Moreover, authors should integrate this information in the wider background referring to sperm motility, since many compounds have been described to affect sperm motility, from viruses’ capside proteins to natural compounds. Is this peptide expected to affect ion channels, mitochondria related metabolism, whatsoever else-…..

Response: The discussion on the methods of screening is very important because in the field of toxinomics, it is less common to use phenotypic screening and even more unusual to start the screening process with a single venom and to be successful in this endeavor. This had to be highlighted. Nevertheless, I completely rewrote the discussion and we concentrate far more on the peptide properties in the context of sperm motility to satisfy the reviewer’s request.

--------------------------------

C7. No English revision has been actually reported also if repeatedly requested

Other comments have been inserted throughout the pdf version

Response: We haven’t received any other PDF version with comments inserted in this second round of revision. With regard to English revisions, we performed all those that were requested by the first review.

Round 3

Reviewer 1 Report

I agree with the authors about their worries to address the interest of the toxicology readers, to whom the manuscript seems to be mainly if not exclusively aimed, with the physiological/bioogical target almost completely missed in the previous versions.

Authors stated that “We illustrate potent activity at 200 nM. This indicates that walterospermin binds with a quite high affinity onto its receptor(s), but the dose-response curve (that should be sigmoidal and not linear) will add no more information, apart from considerably delaying the publication of this manuscript”, without taking into consideration the dose-effect on sperm motility!!!! This is by far more relevant than the sigmoidal rather than non sigmoidal (to be seen) affinity to the receptor.

Venom is SURELY a mixture of different compounds, both inhibiting and activating, as well as both negatively and, maybe, positively affecting the cells, For this reason, comparison between whole and purified waltersperin would have elicited the interest of both toxicology and non-toxicology audience. Above all, dose-dependent effect of the compound would surely have clarified if and how much this compound effectively would have been applied for the in vitro fecundation techniques.,

However, despite the polemical vein of the authors in answering the questions, the revised version of the manuscript addressed many issues, with few others to be elucidated, and authors should emphasize the limit of this study in the end of discussion

Author Response

We thank the reviewer for all the suggestions made during these rounds of review. We obviously agree that they all helped improving the manuscript and we thank him/her for the time devoted to this work.

We have followed the last advice asking to "emphasize the limit of this study in the end of discussion". This is now done at the end of the discussion section (highlighted in yellow). We took this opportunity highlighting also all the experiments that may be performed in the future, that are lacking in the current study, to give value to this peptide and associated receptor on which it is acting. We also mentioned the benefit a dose-response could bring in defining the lowest dose for stimulating sperm motility. We have done our best to satisfy all requests.

With kind regards.